# Scalability and Performance of LiDAR Point Cloud Data Management Systems: A State-of-the-Art Review

**Chamin Nalinda Lokugam Hewage** [1,*], **Debra F. Laefer** [2,3], **Anh-Vu Vo** [1], **Nhien-An Le-Khac** [1] **and Michela Bertolotto** [1]

1    School of Computer Science, University College Dublin, D04 V1W8 Dublin, Ireland
2    Center for Urban Science and Progress, New York University, New York, NY 11201, USA
3    Department of Civil and Urban Engineering, New York University, New York, NY 11201, USA
*    Correspondence: chamin.lokugamhewage@ucdconnect.ie

**Abstract:** Current state-of-the-art point cloud data management (PCDM) systems rely on a variety of parallel architectures and diverse data models. The main objective of these implementations is achieving higher scalability without compromising performance. This paper reviews the scalability and performance of state-of-the-art PCDM systems with respect to both parallel architectures and data models. More specifically, in terms of parallel architectures, shared-memory architecture, shared-disk architecture, and shared-nothing architecture are considered. In terms of data models, relational models, and novel data models (such as wide-column models) are considered. New structured query language (NewSQL) models are considered. The impacts of parallel architectures and data models are discussed with respect to theoretical perspectives and in the context of existing PCDM implementations. Based on the review, a methodical approach for the selection of parallel architectures and data models for highly scalable and performance-efficient PCDM system development is proposed. Finally, notable research gaps in the PCDM literature are presented as possible directions for future research.

**Keywords:** point cloud data; point cloud data management; scalability; performance; parallel architectures; data models

## 1. Introduction

Light detection and ranging (LiDAR) is a technology that uses light, most commonly from a laser, to detect and measure the distance to objects. LiDAR sensors can be deployed on different mapping platforms for topographic mapping purposes. Prominent examples include aerial platforms (e.g., an aircraft, a helicopter, a drone) terrestrial platforms (e.g., a car/boat, or a stationary LiDAR sensor), and most recently, smartphones and tablets. LiDAR mappings produce their outputs as collections of explicitly georeferenced, three-dimensional (3D) point data. Such point data collections are called point clouds (PCs). PCs are important sources for 3D geo-information [1–3]. Each point in a PC represents the x-, y-, and z-coordinates of the sensed object, along with other attributes (e.g., timestamp, laser intensity, etc.) that are dependent on the LiDAR sensor and the presence or absence of integrated global positioning and/or camera instrumentation.

Today, the utilization of LiDAR mapping is increasingly prevalent in many applications. As an example, airborne LiDAR mapping is widely used in large-scale, 3D mapping projects at national, regional, and municipal levels. As of 2020, at least nine countries had completed national LiDAR surveys. They include Denmark, Estonia, Finland, Netherlands, Poland, Spain, Slovenia, and Sweden [4–8]. In the US, aerial laser scanning (ALS) is being conducted at the national level under the 3D Elevation Program (3DEP), which aims to complete the national mapping by 2023. As of the end of the year 2021, the program had documented 84% of the US [9]. Such national ALS projects result in massive volumes of

point cloud data. For example, the US's 3DEP has produced nearly 14 trillion points [10]. In the Netherlands, four nationwide ALS projects (AHN1–AHN4) have been completed. AHN2 exceeded 640 billion point records [8], and future missions are expected to result in larger data sets due to the increasing point densities, which started with as few as 1 point per m$^2$ (in AHN1) to now 20–35 points per m$^2$ (in AHN4). In addition to nation-wide mapping projects, ALS is also widely conducted at regional and municipal levels. Examples include the 2015, high-resolution, ALS survey of a portion of Dublin Ireland, which generated over 1.4 billion data points for an area of 1.5 km$^2$ [11]. Similarly, terrestrial LiDAR platforms have been employed for city and region-scale topographic mapping projects and have collected massive volumes of point data [12–15].

As LiDAR point cloud data are becoming increasingly available, a growing amount of research is being invested in developing efficient systems for point cloud data management (PCDM). While some of those systems adopt file-based techniques to store and query point cloud data, a significant number aspire toward developing PCDM systems that rely on database technology. The crux of PCDM research has aimed to cope with the large volumes of heterogeneous point cloud data. Thus, PCDM solutions that rely on database technology have predominantly focused on achieving greater scalability, while preserving acceptable performance levels. The term "*scalability*" is contextual. In this work, scalability is discussed in terms of developing highly scalable, data-intensive systems or in other words, from the point of data management. Therefore, the term scalability is predominantly focused on how data-intensive systems with fast-growing data demands and query (i.e., traffic) volume demands while ensuring efficient storage and data retrieval times. This definition is obtained from Kleppmann, 2017 [16]. More specifically, the scalable management of point cloud data is being explored in terms of designing and developing data-intensive systems. In other words, PCDM systems are being sought that can accommodate the growth in the different load parameters, while ensuring commensurate performance. These load parameters include high growth in point cloud data volumes, high growth in the number of points retrieved from databases for single user and multiple user scenarios, and high volumes of point cloud data ingestion where necessary for the measured performance metric (e.g., response time taken to retrieve of 1 million points when managing 1 billion points).

To this end, PCDM research has explored various parallel architectures and distinct data models. This paper discusses how different parallel architectures and data models have contributed to scalability and performance. As will be described in Section 2, there have been significant PCDM research studies (in terms of spatial indexes), data encoding techniques, the benefits of adopting databases for PCDM, and data acquisition techniques, there has yet to be a paper that reviews the scalability and performances of existing PCDM systems, particularly combining the impact of both parallel architectures and data models. In addition, the discussion of scalability has been primarily limited to the data model layer. Typically in systems, both architecture and data models are tightly coupled. To consider this tight coupling, this paper discusses aspects of scalability and performance with respect to both parallel architectures and data models.

### 1.1. Contributions

The key objective of this paper is to review the scalability and performance of state-of-the-art LiDAR PCDM systems. The contributions of this paper can be summarized as follows:

1. A novel review of PCDM literature, which analyses the scalability and performance of existing solutions.
2. A thorough discussion of the impact of architectures and data models toward scalability and performance in the context of PCDM.
3. An in-depth analysis of the implementation aspects of PCDM systems.
4. An overview of research experiments published in the PCDM literature, including the synthesis of different queries of interests, data sets, and performance measures

    obtained in the experiments, to demonstrate the spectrum of research conducted in the PCDM area.

5.    A novel, concrete workflow for the selection of parallel architectures and data models for PCDM system development.

6.    A list of notable research gaps in the PCDM literature.

7.    A discussion of the most promising future research directions, including the identification of the need for an agile extensible framework for methodical testing and evaluation of the performance and scalability of PCDM systems.

### 1.2. Paper Organization

The remainder of the paper is organized as follows. Section 2 presents a synopsis of related work in relation to PCDM surveys and reviews and highlights the motivation for writing this paper. Section 3 provides background, including the definitions of scalability and performance. Section 3 also explains parallel architectures and introduces databases and data models that are adopted in state-of-the-art data management systems. Section 4 provides additional technical details on parallel architectures and provides a thorough discussion on scalability aspects and their relationships to PCDM. Similarly, Section 5, investigates various data models and delineates scalability and performance aspects in the context of PCDM. Section 6 furnishes an in-depth analysis of state-of-the-art PCDM systems and a comprehensive list of research activities conducted in PCDM research. Section 7 proposes guidelines on selecting parallel architectures and data models for highly scalable PCDM system development. In addition, Section 7 presents the most pressing research gaps in the PCDM literature. This is followed by the conclusions in Section 8.

### 2. Related Work

While multiple approaches related to PCDM have been presented quite comprehensively in the scientific literature, many excellent past surveys have not fully captured recent and emerging trends in this rapidly evolving field. As such, this paper synthesizes the contributions from existing surveys, original research papers, book chapters, and theses on PCDM. This can be mainly considered in two areas: (i) research on file-based PCDM, and (ii) research on PCDM relying on database technology. These are then discussed with respect to future system selection, with further identification of areas for which major research questions remain.

### 2.1. File-Based PCDM

The work conducted by Otepka et al. [17] was one of the early survey papers in the PCDM context. This survey defined the geo-referenced point cloud data model as a set of points $Pi$, $i = 1, ...., n$ in a 3D Cartesian space that was related to a geospatial reference system (e.g., the universal transverse Mercator). Point cloud features were also described with respect to two classes: (i) fundamental features, and (ii) derived features. Fundamental features are those that are captured in the point cloud measurement/surveying process (e.g., x-, y-, and z-coordinates). In contrast, derived features are generated in the point cloud processing phase (e.g., classification). The authors emphasized the value of keeping the native point cloud, as opposed to a surface model generated from interpolated data (a common practice a decade ago). In the spirit of retaining the native format, the survey discusses viable, state-of-the-art spatial indexes for native point preservation including the kD-tree, octree, and R-tree. This survey further highlighted the different techniques in organizing geometric attributes (i.e., coordinates) and other attributes, such as intensity or color within point cloud files for managing point cloud data. Point cloud data management issues pertaining to individual attribute organizations within point cloud files were also discussed. However, these do not cover database-oriented PCDMs.

The subsequent work by Graham [18] provides an overview of the structure of LiDAR point cloud data and the exploitation and transportation of point records via different encoding formats [e.g., binary, ASCII, and mark-up language based (e.g., XML)]. The

author also provides a detailed description of the LAS data format by the American Society for photogrammetry and remote sensing (ASPRS). The LAS format is the dominant data sharing format for LiDAR data. Graham's descriptions include major highlights of the LAS 1.4 specification, the LAS data format's relation to LiDAR data processing, and how different point properties and LiDAR project properties are reflected within the LAS format. Most importantly, Graham [18] makes a strong case for the use of file-based systems for PCDM, as opposed to database solutions, and challenges the adoption of database solutions. The author asserts that database solutions are attractive, only if random access to LiDAR data is required and further argues that PCDM should be based on file-based storage for high-throughput production operations based on the supposition that the adoption of databases for PCDM is likely to bring more disadvantages than benefits.

### 2.2. PCDMs Relying on Database Technology

Most research on PCDM relying on database technology, including [3,5,19–24], discusses the limitations of file-based PCDMs. These include unreliability in querying a massive number of LiDAR points, inadequate support for ad-hoc queries, poor horizontal and vertical scalability, and lack of support for data integration and data sharing. In addition, the aforementioned papers thoroughly discuss the benefits of adopting database solutions for LiDAR data management: (i) application independence through data isolation, (ii) efficient support for concurrent access to data, (iii) high scalability (e.g., adoption of distributed databases), (iv) I/O optimization, (v) integration of other point cloud data and with imagery data, (vi) the possibility of data retrieval through standardized declarative query languages, (vii) easy administration and security establishment, (viii) design of special forms of spatial data types for point cloud data, and (ix) improved visualization through multiple levels of level-of-detail point cloud data representation.

One of the leading research efforts in point cloud data organization within the database environment is the work presented in [19], which proposes a comprehensive review and benchmarks of multiple PCDM solutions. The benchmarks include two relational database management systems (DBMSs) (i.e., Oracle and PostgreSQL/PostGIS), a columnar data store (i.e., MonetDB), and a file-based approach using LAStools. In addition, those authors investigated the adoption of high-performing, parallel databases for PCDM via the use of the Oracle Exadata Database Machine (OEDBM). Data loading time, data storage, and query response time of each PCDM are measured and presented under three benchmarks (mini, medium, and full) using the AHN2 dataset. The use of three different data set sizes provides a straightforward and concrete means for assessing scalability. Scalability and performance can also be considered in terms of theory (for details see Sections 4 and 5). Nevertheless, in experimental scenarios, a system's scalability is typically established by obtaining performance results for specific workload scenarios.

In that work by van Oosterom et al. [19], the authors primarily investigate scalability with respect to different volumes of data loads. Their analysis includes a comparison of block model, point cloud data organization (where groups of points are stored together as a block) within Oracle and PostgreSQL and a flat table model point cloud organization (where each point is stored in a table row) within Oracle, PostgreSQL, and MonetDB. The authors also explain the use of the PC_PATCH ([25]) data type and SDO_PC and SDO_PC and SDO_PC_BLK ([26]) data types in PostgreSQL and Oracle databases for point data organization.

A more in-depth review of these built-in data types within Oracle and PostgreSQL is presented by Vo et al. [5]. In that work, comprehensive background information on the inception of aerial LiDAR is presented with an emphasis on its growth in scale, resolution, and popularity. Additionally, the authors present a detailed discussion on aerial LiDAR data modeling in file-based and database environments and the application of spatial indexing in aerial LiDAR data as well as how LiDAR point records are partitioned and organized in tile-based indexing, hierarchical indexing, and integrated multiple indexing

structures. Finally, Vo et al. [5] highlighted the lack of database support for LiDAR full-wave form data management (DM).

Additionally, the importance of distributed, non-relational database solutions for PCDM arises increasingly frequently [3,22,27–30]. While the benefits of adopting distributed solutions and non-relational databases have been articulated (i.e., avoiding the single point of failure and better scalability) in [3,24], but published works to date have not reviewed the scalability and performance of current PCDM systems. In particular, what makes PCDM systems scalable and how that is achieved are topics lacking rigorous study. The most notable points highlighted in this section are summarized in Table 1.

**Table 1.** Current reviews on PCDM and their relevance to scalability and performance.

| References | Main Contribution(s) in Terms of Reviewing PCDM | Relevance to Scalability and Performance in PCDM |
|---|---|---|
| [17] | Comprehensive survey on georeferenced PCs and PCDM in a file-based environment | No explicit focus on scalability or performance is discussed |
| [18] | Detailed overview of LiDAR point cloud data, their encoding formats, and LAS specification | No explicit focus on scalability or performance is discussed |
| | | Justifies the use of PCDM in database environment for random retrieval of point cloud data |
| [3,5,20,22–24,31,32] | Limitations of file-based systems and benefits of adopting databases for PCDM | Recognizes scalability as an important element in PCDM |
| | | Does not review scalability or performance of state-of-the-art PCDM systems |
| [19] | Methodology to assess the scalability of PCDM systems | Recognizes scalability as an important element in PCDM |
| | | Does not review scalability or performance of state-of-the-art PCDM systems |
| [5] | In-depth review of the data types for PCDM in Oracle and PostgreSQL is presented | Recognizes scalability as a pivotal element of PCDM |
| | | Does not review scalability or performance of state-of-the-art PCDM systems |
| [3,22,27,33] | Leading works that demonstrate the possibility of PCDM in the context of shared-nothing-architecture oriented non-relational databases | Does not review scalability or performance of state-of-the-art PCDM systems |

Table 1 demonstrates the absence of extensive investigations into the scalability and performance of PCDM systems. This gap is discussed in detail in Sections 4–6. In this review, current state-of-the-art PCDM systems are explored by encompassing both parallel architectures and data models—two fundamental areas that contribute to making data systems scalable and efficient.

## 3. Background

### 3.1. Scalability and Performance of Data-Intensive Systems

Achieving higher scalability without compromising commensurate performance is a key objective in state-of-the-art, data-intensive application development where extensive data growth is a key issue [16]. For data-intensive applications, scalability is defined as the system's ability to cope with increased data load factors in terms of volume, traffic, and complexity. When designing data-intensive applications, the initial step is to identity and describe the aforementioned load factors and their expected growth over time. Scalability is assessed by measuring the performance for a sequence of load experiments. While performance aspects typically vary by domain and application areas, common performance measures relate to storage, querying, and data loading. Storage performance is typically measured by the amount of bytes/kilobytes required for storing the data (e.g., in PCDM bytes per point data record [3]). Data loading performance is assessed by *throughput*. Querying performance is typically measured by either *throughput* or *response time*. In data-intensive applications, vernacular throughput is defined as the number of records processed or the amount of work performed (e.g., data loading) per unit of time such as per second, per minute, etc. For example, in a PCDM context, data loading (i.e., the work) is typically measured as the number of points ingested per second or kilo points per second [3,19,20]. Response time is defined as the query elapsed time. In PCDM, query response time is commonly measured in milliseconds, seconds, or minutes [19,24]. In scenarios where multiple queries are executed, the query response time is often reported as the average query response time [19] or as the percentile= [3,34]. When providing concurrent access to data, depending on the objective, data-intensive systems trade response time over throughput, or vice versa [16,35]; presently in PCDMs, query performance is not evaluated as throughput.

Scalability and performance are two important dimensions in data-intensive systems (e.g., PCDM) development. Scalability and performance are intertwined [36]. Ineffective scalability typically results in poor performance [35]—meaning, in non-scalable systems, performance will typically degrade with demand (i.e., load) [37]. In addition, scalability gives the option for a better performance. While having both scalability and performance at a higher level is desirable, data-intensive systems typically trade performance to gain higher scalability [35,37]. Thus, state-of-the-art data-intensive systems are usually designed with the objective of absorbing higher loads, while achieving acceptable performance.

### 3.2. Scaling Techniques and Parallel Architectures in Data-Intensive Systems

#### 3.2.1. Advent of Parallel Data-Intensive Systems

Traditionally computer systems followed "classical" von Neumann architecture and, thus, adhere to a single-core, single-processor architecture. Normally, the term core is used for single computing units. Thus, historically, computer systems each had only a single processor, and in that processor, there was only a single computing unit. Today, these computing units are also introduced as cores. Furthermore, these cores are contextually synonymous with the central processing units (CPUs) (i.e., a core = CPU ([38])). With the commercial introduction by chip manufacturers of processors with multiple computing units on a single processor chip, multi-core processors became a mainstream element in computer systems. Concurrently, the advent of the techniques, such as threading enabled the simultaneous utilization of processor cores. Consequently, the overall logical computing units (i.e., logical CPUs) available in the overall computer system increased, thereby enabling parallel computation.

To achieve greater scalability and commensurate performance, state-of-the-art computer systems that support large-scale data management (i.e., data-intensive systems) are designed by networking different processing elements. These processing elements encompass computing units such as multi-core processors, multiple processor chips (single core or multi-core), elements such as RAMS, disks, and even fully autonomous processing elements (i.e., *nodes* [39,40]). Thus, modern data-intensive systems have a larger number

of available computing units (i.e both physical and logical CPUs). This enables CPUs in data-intensive systems to interact in parallel with the data that resides in RAMs and/or disks in parallel and then, also in parallel, perform computations corresponding to data management tasks.

3.2.2. Architectures of Data-Intensive Systems (Parallel Data System Architectures)

As stated previously, today's data-intensive systems are parallel systems. These state-of-the-art data-intensive systems are mainly of three kinds: (i) vertically scaled systems (scale-up systems), (ii) horizontally-scaled systems (scale-out systems), and (iii) systems that encompass a pragmatic mix of both [16]. Data-intensive systems can adopt vertical scaling in two ways. The first approach consists of connecting many CPUs, memory chips, and disks via a fast network under a single operating system. These systems yield *shared-memory architecture* based, data-intensive systems. The second approach connects multiple machines, where each machine has its own copy of the operating system, CPU(s), and memory chip(s) but uses a shared-disk to store data, via a fast network. Applications that adopt this approach are characterized as *shared-disk architecture*-based data-intensive systems. Applications that adopt horizontal scaling are characterized as *shared-nothing architecture* based systems [16,39].

*3.3. Use of Databases in Data-Intensive Systems*

The adoption of databases is a common scenario in today's data-intensive applications. These databases are typically deployed in shared-memory, shared-disk, or shared-nothing oriented parallel architectures. They provide mechanisms to manage the system's underlying data, either on disk persistently or in the main memory, in an efficient manner. In contrast to file-based applications, these database-oriented, data-intensive applications maintain data independence from the respective applications (in file-based applications, the file formats are tightly bound to the application).

Currently, there is a myriad of databases from which data-intensive systems benefit. They serve a multitude of purposes and, thus, have different data models and are deployed in various parallel architectures. Data models are formats to receive and organize input data. Common ones include relational models, relational-columnar models, NoSQL models (i.e., key-value model, wide-column model, graph model), array models, and hybrid (multi-model and NewSQL) models. Databases that adhere to relational models can be deployed under all three data-intensive system architectures. NoSQL model databases and hybrid model databases are mainly deployed in shared-nothing architecture environments [16,39]. While array databases can also be deployed in shared-memory architectures, in practice array databases are predominantly used in shared-disk or shared-nothing architectures [41].

All three parallel architectural styles and the database deployments in their respective architectures have the potential to provide the scalability required for data management, while ensuring commensurate performance. Nevertheless, understanding the impact of each parallel architecture on scalability and performance when adopting data-intensive application development (such as PCDM) is vital. In particular, the flexibility, technical versatility, range of scalability, performance characteristics, and limitations inherent to each parallel architecture (when adapting to a given data-intensive task) are important. Such understanding enables the selection of appropriate parallel architecture for the data-intensive task at hand. Similarly, the impact of the data models also needs to be understood to develop data-intensive applications. Thus, Sections 4 and 5 are provided to discuss the impact of architecture and data models on scalability and performance with respect to PCDMs.

**4. Parallel Architectures toward Scalability and Performance**

The state-of-the-art shared-memory systems and shared-disk systems have the potential to cope with increased data volumes and traffic volumes. However, the scalability of these vertically scaled systems is always limited to the capacity of their respective

systems [42,43]. For example, in shared-memory systems, the ability to accommodate increased data volumes and traffic volumes (i.e., scalability) is bounded by the existing capacities of all data processing and storage elements (i.e., available CPUs, memory units, and disks). If shared-memory systems require to be further scaled for more data volume and traffic volume beyond their capacities, these systems must be migrated to a higher-end machine that has more storage capacity, processing capability, and memory [44]. Similarly, in shared-disk architecture, scalability also depends on the specific characteristics of the system. In particular, to support increasing volumes of data, shared-disk systems might also require being migrated into shared-disk machines that have bigger disk capacities.

Nevertheless, as compared to shared-memory systems, shared-disk systems enable addressing data processing requirements (e.g., data reads, data writes, etc.) by adding more processor-memory (or CPU-memory) nodes to the existing system. This increases the data processing capabilities without any system migration. However, another issue to consider in shared-disk systems is that the data writes can be performed against any node. This means that two or more nodes can attempt to write a data record (i.e., a tuple) at the same time. Therefore, to ensure consistency, the management system must either use a disk-based lock table or communicate the intention to lock the tuple to the other nodes in the system [45]. Due to these additional overheads, adding more nodes in shared-disk architecture does not always result in higher scalability in terms of managing large data volumes and traffic volumes [16].

In contrast, when data are managed in shared-nothing architecture-based systems, the storage, processing, and memory requirements for increased data volumes and traffic volumes can be added without any system migration. More specifically, the addition of fully autonomous nodes that encompass processor(s), memory(ies), and disk(s) is easily doable in shared-nothing architecture-based systems. Furthermore, unlike shared-disk architecture systems, adding nodes often accommodates higher scalability prospects. This is because in shared-nothing architecture, each node independently manages its own data. Hence, when writing data to persistent storage, the locking of nodes does not incur in shared-nothing architecture. Thus, in scenarios where data volume demands and traffic volume demands are increasing, shared-nothing architecture is considered an effective approach [16].

Notably, Ozsu et al. [39], observed that given the same level of parallelism, shared-memory architecture-based systems typically yielded the best performance, while shared-nothing architecture-based systems yielded the least performance. In all three considered architectures adding more nodes/cores or more powerful nodes/cores typically results in higher levels of parallelism in the system. As a result, more processors (and CPUs), memory, and disk units are available for data management tasks. Hence, the performance, particularly the data loading and querying, typically improve. Although the addition of nodes/cores generally results in improved performance, in PCDM literature, this can be seen only in shared-nothing architecture-oriented PCDM work. More specifically, the current shared-memory and shared-disk architecture-oriented PCDM systems do not demonstrate the performance improvements that PCDM systems can yield when more core/processor-memory nodes are added to an existing PCDM system [19,20,24,31,46–49]. A potential reason could be the infeasibility of adding more cores/processor-memory nodes to existing systems once the systems are configured.

The PCDM system tested in [33], which is based on shared-nothing architecture, demonstrated that the addition of fully autonomous nodes to an existing system can improve data querying time. Improved performance in query response time and data loading can also be seen in other shared-nothing architecture-oriented spatial data management systems (e.g., Hadoop-GIS [50], VegaGIStore [51]). Nevertheless, the addition of nodes to shared-nothing architecture systems only improves the performance sub-linearly [39]. Furthermore, adding nodes does not guarantee performance improvements, as improvements are achievable only up to a specific number of nodes, after which the performance gains will be outweighed by communication costs [39].

When analyzing the state-of-the-art PCDM research work, most of the database-oriented PCDM research efforts are shown to be based on shared-memory architecture-oriented systems [19,20,24,31,46,47,49,52,53]. Critically, current research work in PCDM literature does not provide straightforward reasons for the adoption of shared-memory architectures. However, the architectures are mainly influenced by the respective databases used in the experiments. More specifically, these PCDM systems are based on traditional relational databases (e.g., Oracle, PostgreSQL/PostGIS, etc.).

To the best of our knowledge, there exists only one shared-disk architecture-based PCDM system (i.e., the Oracle Exadata Database Machine PCDM presented [19]). Similar to shared-memory architecture-based PCDM systems, the reasoning for only one shared-disk architecture-oriented PCDM system cannot be strictly justified. Nevertheless, most of today's prominent, shared-disk architecture-oriented database solutions, such as Oracle Exadata, IBM Parallel Sysplex, and similar solutions by Microsoft and Sybase, are commercial database solutions. Consequently, the adoption of shared-disk architecture-based database solutions may require significant financial investment [45]. This can be identified as a potential reason for the paucity of shared-disk architecture-based PCDMs.

Some recent PCDM systems are based on shared-nothing architecture [3,27,33,34]. These systems experiment with NoSQL databases (using data models employed in PCDM as described in Section 5) and are designed to deploy under shared-nothing architectures. Vis-a-vis relational databases, the advent of NoSQL databases is new to the data management paradigm. Thus, the efforts in adopting shared-nothing architecture-based NoSQL systems appear infrequently in the PCDM literature. All of those cited herein demonstrated scalability with commensurate performance in managing different volumes of data. These data volumes ranged from several million LiDAR points to 640 billion (existing systems are further discussed in Section 6). To date, the scalability of those PCDM systems was tested with existing large or dense aerial point cloud(s) (e.g., AHN2, Dublin LiDAR dataset, etc.) and not in terms of real-time data growth and traffic volume growth.

As large and dense LiDAR datasets are increasingly prevalent, this poses challenges to systems that utilize shared-disk or shared-memory architectures. Therefore, in situations where real-time data and traffic volume growth becomes significant for PCDM projects that repeatedly capture data over long periods, shared-nothing architecture-based PCDM could provide a promising avenue.

Having covered the different scalable architectures and their impact toward scalability and performance, the following conclusions can be made: (1) higher scalability and performance can be achieved through parallelization; (2) all three parallel architectures can achieve better performance by increasing parallelism; and (3) shared-memory architecture is the least scalable with respect to growing data and traffic volumes, while shared-nothing architecture has the greatest scalability prospects. Additionally, the performance of shared-nothing architecture systems can be improved by adding more nodes to the system, performance can be negatively impacted after node saturation.

All three parallel architectures have been explored in PCDM research. In terms of static point cloud data sets, sufficient parallelism could allow any of the three architectures to perform in a scalable manner. Conversely, in scenarios where a PCDM must accommodate periodic data and traffic volume growth, a shared-nothing architecture-based PCDM system provides greater potential for effective scalability. In addition to the parallel architecture, the data model is another element that significantly impacts performance and scalability. The next section discusses data models.

## 5. Data Models toward Scalability and Performance

Data models are profoundly important in software development, including database software systems [16]. A data model provides a means to represent real-world entities and their relationships in a database (e.g., a user and user's name, address, etc., or timestamp and metadata among LiDAR points). The most common data model is the *relational model* proposed by Codd in 1970 [54]. Database systems based on the relational model, referred

to as relational database management systems (RDBMSs), have been successfully used for decades in almost every application. According to the relational model, data are organized into relations (i.e., tables) that have fixed and precisely defined schema [55,56]. Many relational database management systems (RDBMSs) (e.g., Oracle and PostgreSQL) provide object-oriented features. The data model in such systems is referred to as an *object-relational data model*. Traditionally, most relational databases rely on vertical scaling. As a result, their scalability in terms of data volume and traffic volume is limited. Nevertheless, today, all major relational databases offer horizontal scaling deployments [16] and are capable of managing large traffic volumes, as well as large data volumes.

Both the relational model and the object-relational model have been used for PCDM. For example, by storing one point record per row and one point attribute per column in a table, van Oosterom et al. [19] and Psomadaki et al. [31] used the relational model (without object-oriented features) to store point clouds. Such models are referred to as flat models and have been implemented in both Oracle and PostGIS DBMSs. Flat models allow direct access to each individual point within a point cloud by using standard DBMSs functions. However, the scalability of these flat models is limited mainly because of the cost of handling an excessive number of data records (e.g., a very large indexing structure that manages billions of records). In addition to flat models, both Oracle and PostGIS allow point cloud storage using object-relational models (i.e., Oracle's SDO_PC and PostGIS's PCPATCH, referred to as block models). As the name suggests, points are grouped into blocks, each of which is represented as a binary large object (BLOB) stored in a row of the database. Thus, there are fewer data records compared to an equivalent flat model. Specialized data types and operations are defined in those systems to handle point cloud BLOBs. Van Oosterom et al. [19] demonstrated experimentally that the block models were more scalable and generally allowed faster querying compared to the flat models.

Column-oriented RDBMSs are another class of relational database systems that have been considered in the PCDM literature. Unlike typical relational databases that are optimized for storage and retrieval of rows of data, column-oriented RDBMSs are optimized for fast retrieval of columns of data, thus characterized as column-oriented. These columnar RDBMSs are often designed to exploit parallelism in the underlying hardware. MonetDB and Oracle Exadata Database Machine (OEDBM) are two prominent columnar-oriented, massively parallel RDBMS. Examples of MonetDB include [19,47,48]. As of today, OEDBM research for PCDM is scarce [19]. At the storage levels, both MonetDB and OEDBM follow a flat table approach for storing point records within their storage models.

An alternative group of data models is known as NoSQL. Unlike RDBMSs, many NoSQL databases do not require an explicitly pre-defined schema. Instead, data can have arbitrary structures and are implicitly encoded by the application logic. This feature is often known as schemaless, which helps reduce the cost of schema evolution and better supports semi-structured and unstructured data. Examples of NoSQL data models include the key-value model (e.g., Redis), the graph model (e.g., Neo4j), the document model (e.g., MongoDB), and the wide-column model (e.g., HBase, Casandra, Big Table). NoSQL data models offer a wide variety of options for representing different kinds of data. Each model is optimized for a specific application [57]. Compared to the relational model, NoSQL models allow higher schema flexibility [16,57].

The schemaless nature of NoSQL databases has been explored in the context of PCDM research. For example, Boehm et al. [22] modeled point cloud files as document objects in MongoDB (document model), and Vo et al. [3] investigated four different ways to model point clouds in HBase (using a wide column model). In those cases, NoSQL allowed point clouds to be stored without a predefined, fixed schema. Such flexibility is particularly useful when point clouds are derived from multiple sources (e.g., different sensor types), have heterogeneous point attributes, and do not require ingestion and storage according to a pre-defined standard such as the LAS standard. Compared to RDBMSs, NoSQL databases more easily accommodate a higher level of data complexity. In addition, many NoSQL data models were developed specifically for greater scalability and performance than that

achievable by RDBMSs [16,57]. Many NoSQL systems evolved around shared-nothing system architecture and are, thus, inherently highly scalable [39]. Data partitioning, distribution, duplication (to minimize network communication), and strategic use of distributed indices, hashing, and caching are among the techniques that allow many NoSQL systems to achieve high query performance and scalability [57].

Most NoSQL RDBMSs achieve higher performance and scalability by scarifying major standard features in RDBMSs, such as ACID (*A—atomicity*, *C—consistency*, *I-isolation*, and *D—durability*) properties [39]. ACID properties are vital in applications that require low latency, while the state of the database continuously changes [39]. Such applications are commonly seen in online transaction processing systems, such as banking systems and online ticketing systems [16,39]. To guarantee ACID compliance, a database management system (DBMS) may compromise its scalability [39]. While ACID properties are important for many applications, there is yet to be an agreed view on whether ACID compliance is necessary for PCDM. Nevertheless, some research (e.g., in Vo et al. [3] and van Oosterom et al. [19]) has recognized that the majority of point cloud databases do not change states, as point clouds are rarely updated, inserted, or deleted after ingestion. Based on that, Vo et al. [3] argued that ACID compliance is not strictly required for PCDM. As the result, trading ACID compliance for scalability and performance as is being done in many NoSQL systems can be considered acceptable in the context of PCDM. However, the advent of novel passenger vehicles and ubiquitous iPhone devices might pave the way for creating a new set of applications that would require a rethinking of LiDAR data sets as pseudo-static entities, which could influence thinking about the importance of ACID compliance.

Importantly, high scalability and ACID compliance do not have to be mutually exclusive. EarthServer [58] is an example of an ACID-compliant, scalable database system that can handle point clouds. EarthServer is based on the array data model of Rasdaman (raster data manager) [59]. The primary type of spatial data supported by EarthServer is high-dimensional, raster data. Accommodating point cloud data requires conversion into a raster representation for storage in EarthServer. Arguably, this is a less-than-ideal situation, as rasterization strips the point cloud data of much of its richness and, thus, its value. Examples of this value are readily demonstrated in per-point processing applications (e.g., Vo et al. 2021 [60], Vo and Laefer, 2019 [61]). In addition to the particular case of EarthServer, modern SQL databases such as NewSQL belong to a class of relational database systems [62] that provide high scalability of NoSQL systems and strong consistency and usability of relational databases [39]. However, Pavlovic et al. [49] appears to be the only PCDM system that is built atop a NewSQL DBMS. In [49], the authors employ the SAP HANA database—an in-memory column-oriented RDBMS for PCDM.

A critical shortcoming of most performance benchmarking is that results cover only a limited number of existing data models [63–66]. Moreover, when conducting performance assessments across multiple data models, the experimental settings across each data model must be identical. In the context of current PCDM literature, this means that performance benchmarks of object-relational data models and relational columnar models must be conducted under similar experimental settings [19]. Although the adoption of the relational, columnar, NoSQL, and NewSQL models exists in PCDM, the performance benchmark has not, to date, been conducted under the same experimental settings for all models. Thus, all data models in point cloud data are not possible. Nevertheless, according to Oszu et al. [39] and Davoudian et al. [57] NoSQL databases, NewSQL databases, and other novel database technologies enable and exhibit better performance. Hence, obtaining performance results under the same testing environments for different NoSQL, NewSQL, and traditional relational databases would provide more insight into performance-efficient PCDM.

Traditional relational and object-relational models have been well exploited for PCDM. Major RDBMSs (e.g., Oracle and PostGIS) have provided PCDM solutions for many years. In addition, the use of hardware optimized columnar oriented RDBMSs (i.e., MonetDB, Oracle Exadata) is also visible in current PCDM research. Specifically, NoSQL offers a wide variety of alternatives to the traditional relational model. Several NoSQL PCDM

(e.g., [3,22,33,34,67]) systems and one NewSQL PCDM system (i.e., [49]) can be found in the current literature. The non-relational data models allow point clouds to be represented without a rigid schema. Thus, they can better accommodate complex heterogeneous point cloud data sets. In addition, both NoSQL and NewSQL promise higher scalability and performance than that attainable with RDBMSs. Many NoSQL databases achieve scalability and performance by sacrificing features, such as ACID compliance. While PCDM may arguably not require strict ACID compliance, the drawbacks of NoSQL models must be considered when selecting a data model. Unlike NoSQL, NewSQL systems provide high scalability without sacrificing ACID. The selection of a data model for PCDM should be done by considering the actual system requirements. For example, if scalability is more important to the application than consistency, a NoSQL data model may be a suitable choice. In case the point cloud data must be integrated with existing data currently hosted in RDBMSs, a relational or object-relational model may be a more effective option.

## 6. Analysis of State-of-the-Art PCDM Systems

As seen in Sections 4 and 5, when discussing theoretical aspects of scalability and performance, parallel architectures and data models can be considered independent areas. However, when analyzing already developed systems, such a decoupling between architectures and data models is not straightforward. This is because in already implemented data management systems, both the architecture and corresponding databases (which encompass data models) are tightly integrated. Therefore, when investigating already implemented systems, the architectures and data models and their impact on the system need to be considered in unison. For the purpose of clarity, this section analyses the state-of-the-art PCDM systems by organizing the systems into their respective parallel architectures: namely as (i) shared-memory based PCDM systems, (ii) shared-disk based PCDM systems, and (iii) shared-nothing architecture-based PCDM systems. The data models, corresponding databases, and other information related to PCDM systems are presented and discussed in unison with each architecture.

### 6.1. Shared-Memory Based PCDM Systems

Table 2 presents the most popular shared-memory architecture-oriented PCDM systems. According to Table 2, current shared-memory architecture-based PCDM systems have been successfully tested with data sets that range from 74 million LiDAR points to 23 billion LiDAR points. In [19], the authors highlight that, even with extensive parallelism and considerably more powerful hardware, ingesting 640 billion LiDAR points in their experimental setup compromised scaling and generated unaffordable loading times. When compared to the 23 billion points, which was the maximum points testable in a shared-memory PCDM system in [19], 640 billion points can be considered to be significantly larger (26.8 times so). Thus, in scenarios where the point cloud data to be managed is extremely large (i.e., billions of points) and the anticipated data growth is expected to happen in significant factors (e.g., reaching beyond several hundred of billion points), using shared-memory PCDM systems could be difficult to test experimentally, because of its dynamic nature.

Given the rare updating of LiDAR data sets [3], the tremendous data ingestion times that could be required (e.g., several days to weeks) for large point cloud data sets (e.g., several hundred billion points) might still be acceptable, but that the ingestion of trillions LiDAR points, such as those in nationwide LiDAR scans, could further exacerbate the data loading time. As a result, more adverse scaling perspectives can occur in both research and non-research scenarios. In addition, the availability of resources and parallelism in shared-memory systems could also impede managing such data sets.

**Table 2.** shared-memory architecture-based PCDM systems.

| References | Max: Points Tested | Database | Data Model |
|---|---|---|---|
| [46] | 20 billion | PostGIS/ PostgreSQL | Object relational |
| [47,48] | 23 billion | PostGIS/ PostgreSQL | Object relational |
| | | MonetDB | Relational (columnar) |
| [19] | 23 billion | PostGIS/ PostgreSQL | Object relational |
| | | MonetDB | Relational (columnar) |
| | | Oracle | Object relational |
| [20] | 5.2 billion | PostGIS/ PostgreSQL | Object relational |
| [31] | 74 million | Oracle IOT | Object relational |
| [24] | 496.7 million | PostGIS/ PostgreSQL | Object relational |
| [49] | 1 billion | SAP HANA | Relational (columnar) |

According to Table 2, the current shared-memory architecture-based PCDM systems are mainly built atop two types of relational databases: (i) object-relational databases that provide persistent storage and (ii) relational columnar databases that provide in-memory storage. Generally, in-memory databases are favorable when the size of the data to be managed is compatible with the system's main memory size. Therefore, when adopting in-memory databases for PCDM, the system's main memory should be able to provide storage requirements for all available point cloud data. In addition, there should be sufficient main memory for PCDM operations such as data loading and data querying. However, determining a priori the size of the main memory that an in-memory PCDM system should possess is non-trivial. This is because the overall memory consumption requirements are influenced by multiple factors such as the data encoding technique, number of points, index size, number of attributes per point, and memory required for common point cloud operations.

Nevertheless, when sufficient main memory is available for both point cloud data and operations, in-memory systems can yield better performance. For example, according to [19,48], in-memory PCDM solutions (e.g., MonetDB) yield better data loading times compared to other shared-memory architecture-based PCDM systems that use persistent storage. The performance gain is also enhanced by the inherent parallel execution in the MonetDB. However, due to main memory bottlenecks within in-memory databases, querying of large data sizes can result in poor performance and ineffective scalability [48]. For example in [48], the authors note that when querying point cloud data from a MonetDB PCDM system that stores 2.2 billion points, it yields better query performance compared to a PostgreSQL/PostGIS-based PCDM system that stores 2.2 billion points. However, when point data are queried from a MonetDB-based PCDM system which stores 23 billion points, the counterpart PostgreSQL/PostGIS PCDM system that stores 23 billion points yielded significantly better query response times. They, thus, concluded that PostgreSQL/PostGIS-

like databases, which organize point data as a set of blocks in the disk storage, in contrast to individually treating each point in a separate row in the main memory as in the case of MonetDB, are scalable solutions for managing voluminous point cloud data [48].

MonetDB and SAP HANA, the two in-memory databases used in shared-memory architecture-based PCDM, can also be deployed in shared-nothing architecture [39]. To the best of the authors' knowledge, currently, there are no MonetDB or SAP HANA-based horizontally-scaled PCDM systems. As horizontal scaling provides favorable scalability and performance perspectives. This can be a potential future research direction for shared-nothing-architecture, oriented PCDM.

Existing, shared-memory, architecture-oriented, object-relational PCDM systems are primarily based in either Oracle or PostgreSQL/PostGIS databases. Both adhere to a fixed schema. Therefore, from a scalability point of view, these PCDM systems will be unable to support the heterogeneity of disparate point cloud data sets acquired from different sensors. However, this limitation can be mitigated, if point cloud data are ingested and stored in database tables in which the schema is defined according to the LAS standard.

From a theoretical perspective, shared-memory architecture and relational data models do not provide sufficient support to accommodate growing data volumes, traffic volumes, and data complexities in PCDM. Beyond the research listed in Table 2, some PCDM researchers still adopt shared-memory architecture and relational (object-relational) data models. For example, the need for a dedicated spatial data type for point cloud data, as advocated for in [19], and further examined in [68,69] in a shared-memory architecture-based PCDM system. Work by both [68,69] are built atop Oracle databases and mainly focus on identifying new shared-memory indexing strategies for PCDM. Similar attempts can be also found in [20,31,46,52]. From scalability and performance perspectives, spatial indexing plays a pivotal role. For example, when managing large volumes of point cloud data, efficient filtering or retrieval of required LiDAR points is critical. Thus, spatial indexes that reduce the number of disk reads or decrease the load to the database are attractive due to their positive impacts on query throughput and query response time as performance is improved. As indexing strategies employed in shared-memory architecture PCDM systems are already reviewed quite comprehensively in [5], they will not be discussed further in this paper.

### 6.2. Shared-Disk Oriented PCDM Systems

The Oracle Exadata PCDM system (see [19]) appears to be the only system that adopts shared-disk architecture for PCDM and has been tested beyond 23 billion points. In [19], the authors tested the OEDBM PCDM system with 640 billion LiDAR data points without experiencing adverse data loading or processing times. This example showed the ability of a shared-disk architecture to handle unprecedented data quantities compared to shared-memory architectures. Nevertheless, this is predominantly attributed to the exhaustive memory usage and optimized hardware parallelization inherent to OEDBM. Therefore, presently in PCDM literature, assessing scalability and performance of shared-disk architecture-based PCDM against shared-memory architecture based PCDM is still an open research question.

In both shared-memory architecture and shared-nothing architecture based PCDM systems, the emphasis on indexing to assist point cloud data retrieval is a widely discussed topic. However, in [19], no index was implemented for point cloud data retrieval. This is because OEDBM does not implement additional data structures or indexes for data retrieval purposes. Instead, for the purpose of data retrieval and to reduce disk I/O, OEDBM employs the concept of *reverse indexes*. Reverse indexing is typically achieved via metadata management. Through the use of metadata such as minimum values and maximum values in different columns, OEDBM can perform data retrieval [70] without manual intervention.

In addition, similar to most shared-memory architecture-based PCDM systems, OEDBM is primarily based on the relational model (see Section 5; OEDBM uses a column-oriented

relational model). Therefore, when point cloud data are managed in OEDBM, the data must conform to a fixed schema. Consequently, the scalability with respect to data complexity is limited.

In terms of its computational techniques, OEDBM leverages Message Passing Interface (MPI) library for distributed multi-node computations. MPI is primarily employed in systems that have many nodes and where a high level of synchronization among computational tasks [38] is required. This is achieved through the efficient use of available parallelism in the underlying hardware. Thus, adopting OEDBM for PCDM can provide the benefit of fully exploiting the parallelism in the underlying hardware. However, this requires extensive expertise in efficiently splitting the tasks (i.e., data loading, data querying, etc.) in the available hardware. Consequently, this added complexity could act as a potential barrier for highly scalable PCDM research experiments.

From the above discussion of PCDM systems in shared-memory and shared-disk architectures, these systems have been tested successfully to a maximum of 23 billion and 640 billion LiDAR points, respectively. However, as described in Sections 4 and 5, both shared-memory and shared-disk systems provide restrictive scalability compared to shared-nothing architecture-based systems. Therefore, investigating the scalability of shared-nothing architecture-based, PCDM systems is a logical next step and appears as the next sub-section.

### 6.3. Shared-Nothing Architecture-Oriented PCDM Systems

One of the first efforts in PCDM-based shared-nothing architecture is discussed in [22]. The authors loaded and managed 448 billion LiDAR points using MongoDB, a document-oriented NoSQL database. In that experiment, the large LiDAR files were partitioned into a collection of smaller files. Subsequently, the smaller files were stored within MongoDB collections deployed atop a GridFS file system. In that work, the authors claimed high scalability in terms of large data volumes, because that system could adopt Hadoop Distributed File System (HDFS) for distributed storage, scalability, and higher capacity for voluminous data management [71] and support MapReduce [72], a parallel data processing framework. Nevertheless, the proposed solution is limited to file selection, as the data are managed at the file system level.

Li et al. [29] proposed a general framework for LiDAR PCDM, which was also built atop HDFS and MapReduce but only tested to 4.17 billion points (a fraction of that previously tested with shared-memory architecture-based systems and shared-disk architecture-based systems). The authors' main motivations were the massive storage and data processing demands inherent to PCDM. To that end, spatially organized LAS files were stored within HDFS, and a MySQL database was used to index the LAS files. Their solution also integrated LAStools in every node to generate compatibility with the MapReduce framework to enable LAS file manipulation and support for machine learning efforts.

While [22,29] were built for shared-nothing architecture, the solutions follow a hybrid approach where both LAS file format and database solutions coexist within the same system. Therefore, complete data independence, which is a major objective for adopting databases for PCDM is not guaranteed. Hence, the remainder of this review focuses exclusively on pure database storage solutions.

#### Database-Oriented Shared-Nothing Architecture-Based PCDM

Some of the most important work on shared-nothing architecture-oriented PCDM include [3,33,34]. From a theoretical perspective, these systems are highly scalable for large volumes of data. However, these PCDM systems that were built atop shared-nothing architecture, were only evaluated for a maximum of 1.4 billion points. More specifically, References [3,33,34] were tested only for a maximum of 1.4 billion, 273 million, and 812 million points, respectively, but clearly demonstrate the potential scalability of shared-nothing architecture built PCDM systems. In certain cases, horizontally-scaled PCDM have yielded better scalability and performance compared to shared-memory architecture-

oriented PostgreSQL/PostGIS PCDM systems. This is attributable to the greater availability of resources in the shared-nothing architecture.

In addition to the aforementioned shared-nothing architecture-based PCDM work, GeoWave [67], EarthServer/RASDAMAN [59,73,74], and TileDB [75] also provide support for PCDM in shared-nothing architecture settings. Table 3 provides an overview of the existing PCDM solutions that follow the shared-nothing architecture. According to Table 3, PCDM systems that adopt shared-nothing architecture are mainly based on the wide-column or array model. Furthermore, leveraging MPI-based, highly synchronized computations is mainly employed in array model based, PCDM systems. These systems (i.e., TileDB and EarthServer/RASDAMAN) use HDFS/S3 and PostgreSQL, respectively, as their data storage medium. In contrast, wide column-based PCDM systems use HBase and Accumulo databases as their databases. Furthermore, in contrast to MPI, these HBase and Accumulo-based PCDM systems leverage the Hadoop/MapReduce or Hadoop/Spark data processing frameworks, as their multi-node computation frameworks.

**Table 3.** Shared-nothing architecture-based PCDM systems.

| Reference | Database/ Storage Medium | Data Model | Storage Engine's Access Method | Index Implemented | Computing Technique |
|---|---|---|---|---|---|
| [3] | HBase | Wide-column | LSM-tree | Single Hilbert SFC and Dual Hilbert SFC | Hadoop and MapReduce |
| [33] | HBase | Wide-column | LSM-tree | Z-order SFC | Hadoop and Spark |
| [34] | HBase | Wide-column | LSM-tree | timestamp as the row key (i.e., the index) | Hadoop and MapReduce |
| GeoWave ** | Accumulo | Wide-column | LSM-tree | Hilbert SFC, Z-order SFC | Hadoop and MapReduce |
| EarthServer/ RASDAMAN ** | PostGIS/ PostgreSQL | Array | B-tree | R+-tree, Directory index, Regular computed index | MPI |
| TileDB ** | HDFS/ S3 | Array | B-tree | R-tree | MPI |

To the best of authors' knowledge, there's no openly published research work that reviews or analyses the scalability and/or performance of some shared-nothing architecture-based PCDM systems. These systems are marked with ** in Table 3.

Compared to MPI, the computations in Hadoop/MapReduce and Hadoop/Spark are not highly synchronous. Importantly, Hadoop/MapReduce and Hadoop/Spark provide higher levels of programming abstractions that circumvent programmers from needing skilled knowledge of the underlying hardware. Nevertheless, knowledge of the underlying hardware will enable Hadoop/MapReduce and Hadoop/Spark programs to more efficiently utilize the underlying hardware parallelism. Therefore, from a pragmatic perspective, the use of Hadoop/MapReduce and Hadoop/Spark for shared-nothing architecture-based PCDM could be considered less complicated to PCDM researchers compared to MPI-based PCDM.

As seen in Section 6.2, MPI can be leveraged atop relational models (e.g., OEDBM in shared-disk architecture). However, according to Table 3, MPI and relational model-based PCDMs are not popular. Therefore, the adoption of MPI, relational data models in

horizontally-scaled architectures could be a potentially fruitful direction for scalable PCDM investigation, particularly where scalability with respect to data complexity is trivial.

As described in Sections 5 and 6.2, the adoption of an array model requires the loss of a point cloud-native format. Yet as previously noted in Section 5, point cloud data must be considered as the primary data set where its preservation along with connections to any post-processing is to be retained. The value of such a mindset has already been demonstrated in the context of solar potential analysis [61], where interim outputs were the final outputs needed for shadow analysis [30]. In an array model, the native point cloud format is transformed into a raster format. Thus, coping with data complexities in systems such as EarthServer/RASDAMAN and TileDB are prohibitive, even though they can be horizontally scaled to multiple, independent nodes to efficiently manage the needed data volumes and traffic volumes.

Notably, since both HBase and Accumulo NoSQL databases relax the requirements to adhere to a rigid schema, systems that are built atop HBase and Accumulo could offer greater scalability accommodating data complexities. Hence, [69] argues that managing point cloud data in HBase or Accumulo-based systems, such as GeoWave, would facilitate requirements, such as assemblage of heterogeneous point cloud data. Furthermore, systems that are built atop HBase and Accumulo provide flexibility to scale to a large number of nodes. Therefore, from a theoretical perspective, systems that adopt HBase or Accumulo, such as wide-column databases for PCDM could achieve greater scalability with respect to data volume and traffic volume, as well as data complexity.

On another note, key-value databases and wide-column databases are optimized for write-intensive workloads [76]. Such databases typically are constructed atop log-structured merge tree (LSM tree)-based storage engines vis-a-vis read-optimized storage engines that are typically built atop B-tree structures [16]. Commonly, once point cloud data are ingested, a PCDM system is expected to perform predominantly read-intensive tasks. Thus, the adoption of write-intensive databases can be seen as a contradiction for PCDM in system design objectives. Nevertheless, in scenarios where read optimization is expected in LSM tree-based databases, such as HBase, Bloom Filter adjustments are doable [77]. However, the existing PCDM research that employs HBase has not explored the potential gains that could be obtained from Bloom Filter adjustments.

Similar to shared-memory architecture-based PCDM systems, shared-nothing architecture-based PCDM systems also invest efforts in implementing efficient spatial indexing strategies for point cloud data retrieval. Table 3 shows that except for TileDB, [34] and EarthServer/RASDAMAN systems, other research work that uses HBase or Accumulo, like databases have implemented spatial indexes atop point cloud data. Such efforts have been tested predominantly with two variations of space-filling curves (SFCs): (i) Hilbert SFC and (ii) Morton curve. Attempts at implementing other space-based point access techniques such as hierarchical indexes (e.g., quadtree) for PCDM in wide-column databases do not appear in the peer-reviewed PCDM literature. Indeed, there could be challenges in implementing and managing tree-like, hierarchical data structures in shared-nothing architecture context. However, there have been attempts at quadtree implementation atop an HBase in MD-HBase in which the authors demonstrate the use of an on-disk, persistent quadtree data structure in performing 2D point data retrieval in a series of range and $k$ nearest neighbor queries [78]. Thus, the implementation of a quadtree, such as a data structure atop wide-column databases for the purpose of PCDM can be seen as a potential future research direction, especially for spatial queries with 3D point cloud data.

The adoption of shared-nothing architecture is also extended to full waveform (FWF)-LiDAR data management. As opposed to discrete points in point cloud data, FWF-LiDAR comprises the full waveform version of the raw signal. This includes the *pulse* component, i.e., the line segment that represents the location and orientation of the laser beam, the *wave* component, i.e., the series of signal magnitude values, and the *point* component which is the derivation of the processing of pulse and wave components. This richer information in FWF-LiDAR provides greater insight into scanned scenes, yet demands more storage and

computing power. In response, Vo et al. [79] developed a novel spatial–temporal indexing technique and scalable data management solutions for FWF-LiDAR data by adopting the HBase database. The spatial index strategy of Vo et al. [79] employs a six-dimensional Hilbert index. This index is based on the two edges of the *pulse* component of FWF-LiDAR: the x-, y-, and z-coordinates of the two edges of the *pulse* component. The temporal indexing is based on the flight line id, which is unique to each pulse component in the FWF-LiDAR data.

As described in Section 4, the addition of nodes is a means to improve performance when adopting shared-nothing architecture. For example, in the work [51,78,80,81], the addition of nodes resulted in improved performance for data querying and data loading, in the respective spatial data management systems. Nevertheless, within the context of PCDM, to date, only [33] provides sufficient evidence toward this potential under 3, 5, and 9 nodes. Although this experiment clearly depicts the benefit of shared-nothing architecture in performance improvement, more systematic testing is needed with multiple data sets, various standard queries, and other expected actions such as data loading and concurrent data querying.

### 6.4. Comparison of Scalability and Performance of Current PCDM Systems

In this section, a comprehensive overview of underlying architectures and data models and other related pivotal information on current PCDM solutions are presented, along with their scalability potential and prominent performance aspects with respect to storage, querying, and loading.

Achieving scalability, while ensuring acceptable performance, is the ultimate goal in PCDM research. Almost all PCDM experiments to date demonstrate some scalability with respect to data volumes but are not easily compared because of the use of different hardware, various data sets, and disparate queries. Arguably, the most reliable way to experimentally compare systems is to conduct systematic benchmarks similar to that done by van Oosterom et al. [19]. Such an approach enables the direct comparison of systems with different data models and distinct indexing techniques. The absence of that type of experiment to date greatly complicates meaningful comparisons of PCDM systems. Despite these understood limitations, evaluation of existing comparisons can provide insights. Thus, Table 4 is provided.

**Table 4.** Data loading, data storage, and different range query performance time achieved in current PCDM solutions.

| Ref: | Database | Storage Model | Index Strategy | Queries of Interest | | kNN | Point | temp: | Dataset (by Project/City/ Country) | Max: Points Tested | Data Loading (bytes/s) | Data Storage (bytes/point) | Query Response Times (s) | Points Returned |
|------|----------|---------------|----------------|------|------|------|-------|-------|--------------------------|--------------------|------------------------|----------------------------|--------------------------|-----------------|
| | | | | **Window** | | | | | | | | | | |
| | | | | **2D** | **3D** | | | | | | | | | |
| [19] | PostgreSQL/ PostGIS | flat | B-tree (x,y) | ✓ | | | ✓ | | AHN2 | 23 B | 243,022 | 77.1 | 18.02 | 718,131 |
| [19] | PostgreSQL/ PostGIS | block | NA | ✓ | | | ✓ | | AHN2 | 23 B | 1,882,931 | 4.6 | 2.19 | 718,131 |
| [20] | PostgreSQL/ PostGIS | block | R-tree, B-tree | ✓ | | | | | Vosges | 5.2 B | 125,000 | 7.5 | NA | NA |
| [20] | PostgreSQL/ PostGIS | block | R-tree, B-tree | ✓ | | | | | Paris | 2.15 B | 74,500 | 26.9 | 0.6 | 1,200,000 |
| [46] | PostgreSQL/ PostGIS | block | 2D tile index with metadata | ✓ | | | | ✓ | Italy, Austria | 20 B | 30,000 | 190 | NA | NA |
| [19] | Oracle | flat | B-tree | ✓ | | | | | AHN2 | 23 B | 1,459,065 | 43.2 | 18.2 | 718,021 |
| [19] | Oracle | block | Hilbert-R-tree | ✓ | | | | | AHN2 | 23 B | 119,881 | 9.5 | 1.3 | 718,131 |
| [31] | Oracle | flat | Hilbert (x,y) | ✓ | | | | | AHN2 | 73 M | 278,427 | 25 | 0.33 | 3927 |
| [31] | Oracle | flat | Hilbert (x,y,z) | ✓ | | | | | AHN2 | 73 M | 110,570 | 20.2 | 1.12 | 3927 |
| [31] | Oracle | flat | Hilbert (x,y,t) | ✓ | | | | | AHN2 | 73 M | 107,444 | 25.7 | 0.06 | 3927 |
| [31] | Oracle | flat | Hilbert (x,y,z,t) | ✓ | | | | | AHN2 | 73 M | 92,420 | 20.2 | 0.11 | 3927 |
| [68] | Oracle | flat | Morton (x,y) | | | NA | | | AHN2 | 273 M | 380,011 | 51.1 | NA | NA |
| [68] | Oracle | flat | SDO_ Point(x,y,z), scale(LoD) | | | NA | | | AHN2 | 273 M | 123,247 | 153 | NA | NA |
| [19] | MonetDB | flat | Imprints | ✓ | | | ✓ | | AHN2 | 23 B | 2,719,439 | 22.9 | 16.74 | 718,021 |
| [52] | MonetDB | flat | Imprints | ✓ | | | | | AHN2 | 23 B | 5,888,822 | 22.9 | 0.32 | 718,021 |

**Table 4.** *Cont.*

| Ref: | Database | Storage Model | Index Strategy | Queries of Interest | | | | | Dataset (by Project/City/ Country) | Max: Points Tested | Data Loading (bytes/s) | Data Storage (bytes/point) | Query Response Times (s) | Points Returned |
|------|----------|---------------|----------------|---------------------|---|---|---|---|---------------------|--------------------|------------------------|----------------------------|--------------------------|-----------------|
| | | | | Window | | *kNN* | Point | temp: | | | | | | |
| | | | | 2D | 3D | | | | | | | | | |
| [52] | MonetDB | flat | Morton-added | ✓ | | | | | AHN2 | 23 B | 415,725 | 30 | 0.2 | 718,021 |
| [52] | MonetDB | flat | Morton-replaceXY | ✓ | | | | | AHN2 | 23 B | 592,492 | 15 | 0.3 | 718,021 |
| [19] | OEDBM | flat | no indexes | ✓ | | | | | AHN2 | 21B | 22,706,269 | 4.5 | 0.59 | 369,352 |
| [3] | HBase | flat | Hilbert | | | ✓ | ✓ | | Dublin | 1.4 B | 41,283 | 235.5 | 0.05 | NA |
| [3] | HBase | flat | Hilbert | | | ✓ | ✓ | | Dublin | 1.4 B | 181,110 | 48.3 | 0.04 | NA |
| [3] | HBase | block | Hilbert | | | ✓ | ✓ | | Dublin | 1.4 B | 1,344,047 | 31.2 | 0.07 | NA |
| [3] | HBase | block | Hilbert | | | ✓ | ✓ | | Dublin | 1.4 B | 2,372,243 | 26.9 | 0.08 | NA |
| [34] | HBase | flat | NA (time-stamp) | | | | | ✓ | Dublin | 812 M | 259,011 | 12.3 | 48 | 13,498,454 |

'NA' stands for 'not applicable' and/or 'not available', 'B' stands for 'billions' and 'M' stands for 'millions'.

Inspect Performance Results of Current PCDM Systems

Table 4 and Figure 1 summarize the performance results (i.e., storage performance, data loading performance, and window query performance) reported to date for *prominent* PCDM systems. The focus is on window queries, as they are commonly used across all PCDM experiments, although with some differences. For example, Ref. [19] is primarily based on 2D window queries and kNN queries, whereas [3] employs 3D window queries. As reporting all experimental results in one table requires extensive consolidation, specific considerations were made while constructing Table 4. These are as follows:

- In scenarios where different data set sizes were tested, the results obtained from the largest are reported herein.
- Additional information such as the number of threads used for loading, nature of the window query, spatial index used, and storage model implemented are provided where appropriate.
- Data storage is mainly ascertained by considering the total disk/memory usage, i.e., in experiments where the size of the index and data are provided separately, the summation of the two (i.e., total storage consumption) is reported.
- While the results reported are impacted by data heterogeneity, there is not an easy means of characterization and, thus, must be considered as an uncertainty in the reporting.

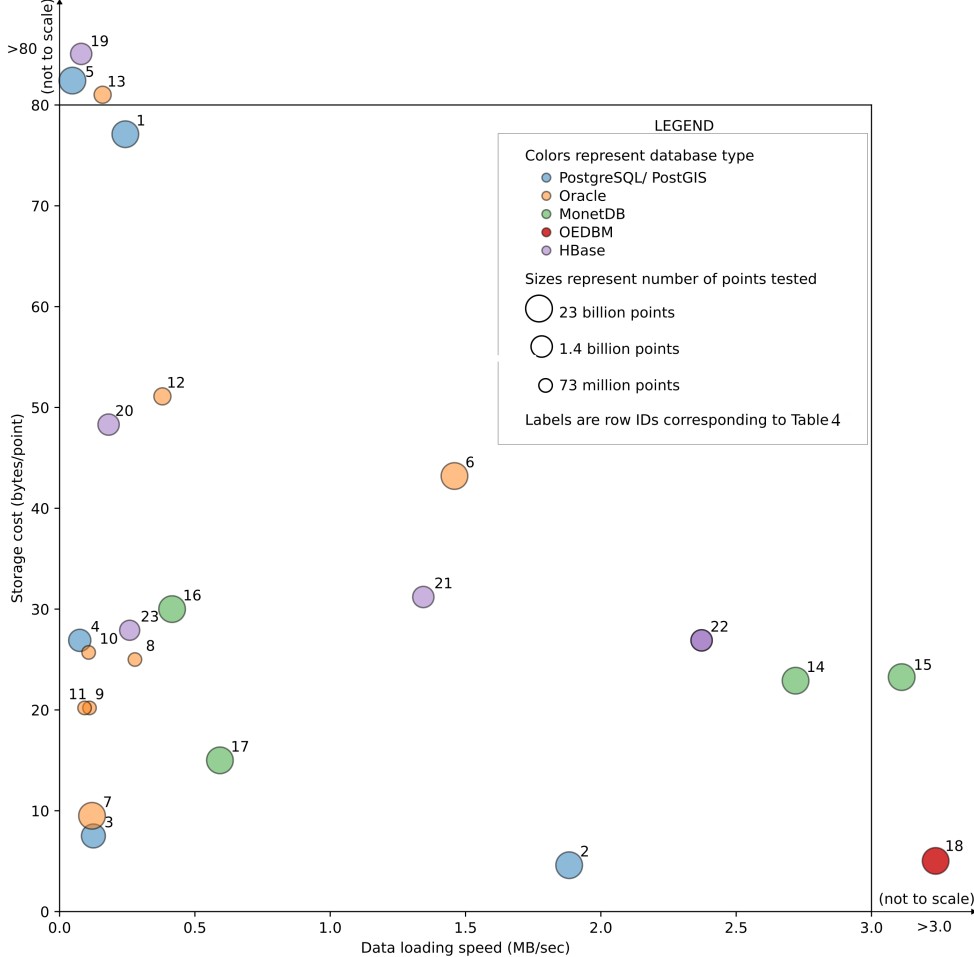

**Figure 1.** Data loading speeds and storage costs of the existing PCDM solution.

With respect to Figure 1, data loading speed and storage costs of several PCDM systems were significantly higher (i.e., an outlier). Thus, preserving space and clarity in Figure 1, outlier data points are depicted outside the main plot region.

Based on Table 4 and Figure 1 with respect to current experimental scenarios, the following observations can be made:

- The OEDBM PCDM system—a multi-node shared-disk architecture-based relational database-oriented PCDM system had the best (highest) reported points/seconds ratio.
- The OEDBM PCDM system also had the best (lowest) bytes/points ratio (i.e., the number of bytes per point), with the aid of an in-built compression mechanism (i.e., query high compression mode).
- Among the PostgreSQL/PostGIS-oriented PCDM systems:
    - The block model implemented in [19] yielded the lowest bytes/points ratio (using blocks of 3000 points).
    - This system also yielded the highest points/loading time and ratio among the PostgreSQL/PostGIS systems.
- Among the Oracle database-oriented PCDM systems:
    - The Oracle flat model implemented in [19] yielded the highest points/loading time ratio.
    - The Oracle system that yielded the lowest bytes/points ratio was the Oracle block model implemented in [19].
- Amongst the MonetDB-based systems:
    - The *Morton-relaceXY* implementation yielded better bytes/points ratio [47].
    - The *Imprints index* implementation yielded a better points/loading time ratio [47].
- Investigation of the 2D window queries is more common compared to the investigation of the 3D range or kNN queries for LiDAR point cloud data.
- Data loading and concurrent data querying can be done in parallel. However, many researchers have not explicitly explored parallelism under data loading and querying.

In the process of amassing data loading results, querying results, and storage results for Table 4, a few notable gaps in the reporting of shared-nothing architecture-based PCDM research experiments were identified that should appear in future reporting. These include (i) the replication factor in the database (e.g., in HBase, the default replication factor is 3 [82]—meaning: that three copies of data are stored in the database); (ii) the number of nodes used in the data loading process; (iii) communication costs incurred in data retrieval (which is an overhead); (iv) index size; and (v) point cloud data distribution across nodes.

Such information is vital. For example, according to [3], PostgreSQL/PostGIS consumed 21.0 bytes to store a LiDAR point record whereas the HBase model-4 consumed 28.4 bytes per point. Assuming that the default replication factor 3 is configured in [3]'s HBase database, it could be argued that 28.4 bytes value represents storage requirement for 3 points, not one point. If so, the storage performance that [3] obtained for its best-case scenario is nearly 100% more efficient as compared to PostreSQL/PostGIS system. However, since that information is not available, such a conclusion cannot be made without further experimentation. Similarly, information, such as the number of nodes participating in the data loading process, the size of the index, and the time incurred for data communication are essential for an insightful analysis of scalability and performance of shared-nothing architecture-based PCDM systems.

Current state-of-the-art PCDM systems are implemented atop parallel architectures. Published experiments to date demonstrate that shared-memory architecture-based PCDM could result in poor scalability and performance with large data volumes. In addition, current shared-memory architecture-based PCDM systems cannot support heterogeneity of data, as the systems are based on a relational data model. Similarly, shared-disk architecture-based PCDM systems demonstrate limited scalability when in the presence of data complexities. Nevertheless, in current PCDM literature, shared-disk architecture-based PCDM systems are the only database-oriented PCDM systems that have been tested with extremely large data sets (i.e., up to 640 billion points); compared to the maximum of 1.4 billion points tested on the current shared-nothing architecture-based PCDM system.

In terms of supporting data complexities, shared-nothing architecture-based systems that utilize HBase and Accumulo, demonstrate much promise. Scalability with respect to concurrent data access or traffic scalability is highlighted as an important aspect of PCDM [19]. However, PCDM scalability (with respect to different traffic volumes) is not well researched, queries to date have been 2D window queries, with substantially fewer attempts with 3D window and kNN queries.

## 7. Discussion

Discussing issues that arise in developing new scalable, performance-efficient PCDM systems are critical, as well as highlighting crucial gaps in the PCDM research domain. Therefore, this section provides guidance on selecting parallel architectures and data models when specific requirements arise. Future research avenues are also presented.

### 7.1. Guide to Selecting Parallel Architectures and Data Models for PCDM

Figure 2 depicts the avenues available for highly scalable, performance-efficient PCDM systems implementation. From the parallel architecture's viewpoint, shared-nothing architectures provide more flexibility for the choice of data models compared to both shared-memory and shared-disk architectures. Similarly, from a data model point of view, relational data models (i.e., including object-relational, relational columnar, etc.) provide more flexibility, as they can be deployed in any parallel architecture. Furthermore, if the choice of parallel architecture is primarily on shared-memory architectures, the data model choices available will be relational models and NewSQL models. On the other hand, shared-disk architecture is limited to array and relational models.

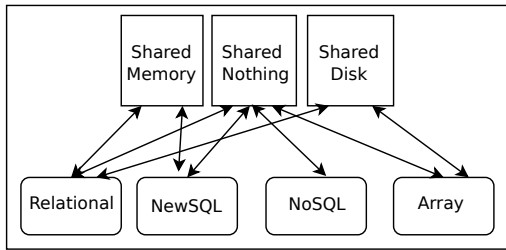

**Figure 2.** Available choices in adopting parallel architectures and data models for PCDM systems development.

Although, as previously noted, NoSQL models inherently provide higher scalability and performance compared to relational models, NoSQL models are primarily designed to deploy in shared-nothing architectures. Thus, the adoption of wide-column and/or key-value databases will require PCDM system developers to adhere to shared-nothing architectures, which may unduly restrict certain types of querying.

Figure 2 provides some broad guidelines for the choice of parallel architecture and data model for PCDM. However, based on the review provided herein, the adoption of specific parallel architectures and data models for PCDM requires meticulous attention to ensure acceptable performance and future usability in terms of data size and complexity. In particular, the choice of parallel architecture and data model should be made through careful investigation with respect to the specific requirements of the intended PCDM system. From a decision-making point of view, such requirements can be mainly categorized into three areas, namely: (i) anticipated changes in point cloud data volume, (ii) anticipated changes to the traffic volume, and (iii) other PCDM system requirements (e.g., requirements associated with the data models/databases and inherent characteristics of point cloud data).

Figures 3 and 4 provide a workflow for decision-making with respect to anticipated changes to traffic volume and point cloud data volumes. From the review, all parallel architectures are capable of providing scalability and commensurate performance for PCDM. Nevertheless, as highlighted in the review, the scalability and performance of the PCDM systems have been tested over precisely defined data sizes. Consequently, the

maximum data size to be managed in each experiment is known a priori. In such cases, the adoption of any parallel architecture is possible for PCDM. This flexibility is allowed because the required storage and data processing capabilities can be designed explicitly into the parallel architectures, and may permit consideration of a wider range of data models (as depicted in Figure 2).

Where anticipated, data and/or traffic volumes are unknown, but significant changes are expected, shared-nothing parallel architectures seem more suitable. Indeed, as depicted in both Figures 3 and 4, the capacities of shared-memory or shared-disk parallel architectures could act as potential barriers.

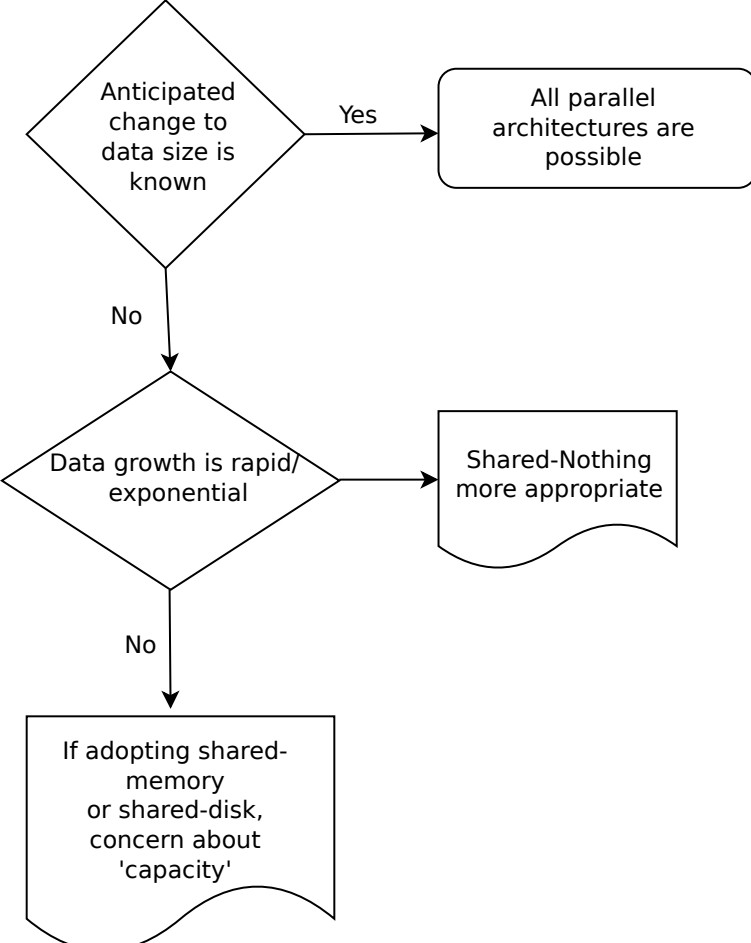

**Figure 3.** Choosing parallel architectures based on anticipated changes to data volume.

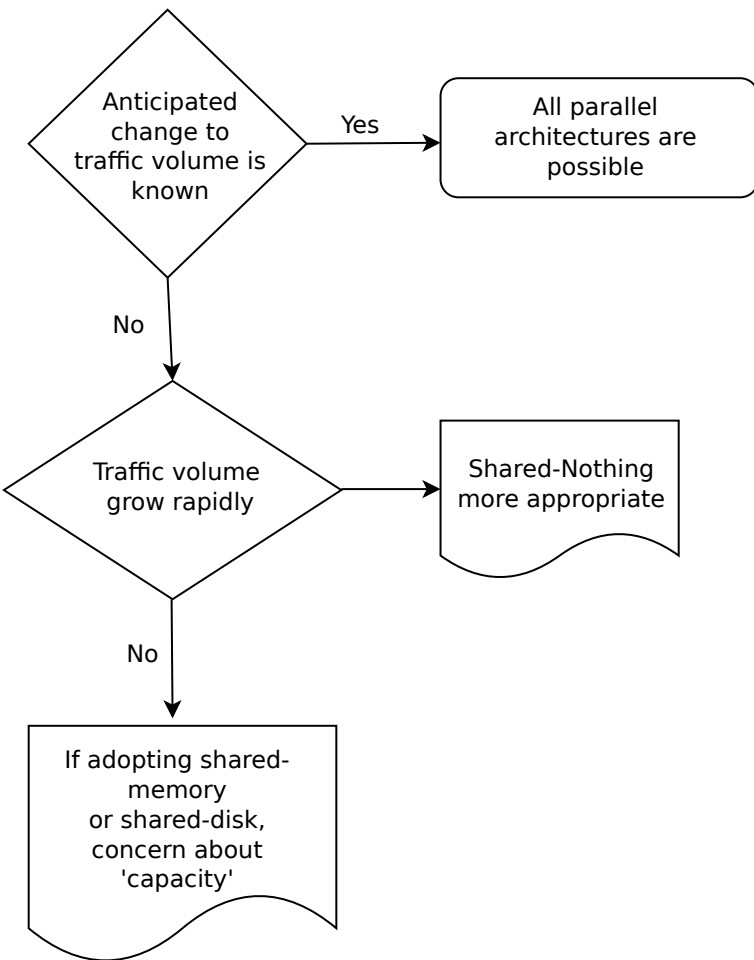

**Figure 4.** Choosing parallel architectures based on anticipated changes to traffic volume.

While intuition with respect to parallel architecture could be based on anticipated changes in point cloud data volumes and traffic volume demands in a PCDM system, other system requirements associated could provide better insights for selecting the most appropriate data models. To this end, Table 5 provides an overview of the data models available for different requirements specific to PCDM system development (Please note that Table 5 has a column name titled: Wide-column, instead of NoSQL. The reason is that, as of the current juncture in PCDM, wide-column is the only NoSQL model that is experimented with in the full database-oriented PCDM). For example, if higher scalability and performance are integral requirements, NewSQL models and wide-column NoSQL models may provide competitive solutions. Indeed, although array data models are scalable, to the best of our knowledge, the scalability of array databases has not directly been compared with NewSQL and NoSQL models. Similarly, if ACID requirements and/or fixed schema requirements and/or the relationships among point cloud data or data sets are a priority, relational, NewSQL, and array models may provide more viable solutions. On the other hand, if requirements such as the assemblage of heterogeneous point cloud data are a priority, wide-column-oriented NoSQL data models should be considered. The fundamental level guidelines depicted in Figures 2–4 and Table 5 will enable strategic decision-making in the development of highly scalable performance-efficient PCDM systems.

**Table 5.** PCDM requirements that can cater through data models.

| Requirement/Characteristics | Data Models | | | |
| --- | --- | --- | --- | --- |
| | Relational | NewSQL | Wide-Column | Array |
| Assemblage of heterogeneous point cloud data/ (Schemaless storage) | | | ✓ | |
| ACID requirement | ✓ | ✓ | | ✓ |
| High scalability and performance are inherent characteristics of the data model | | ✓ | ✓ | ** |
| Fixed schema | ✓ | ✓ | | ✓ |
| Relationship among point cloud data /datasets (one-one, one-many, many-many) | ✓ | ✓ | | ✓ |

As stated in the main text, the scalability of array databases has not directly been compared with NewSQL and NoSQL models. Thus, the cell value in the table is indicated as **.

### 7.2. Further Research Avenues

In addition to the gaps stated with respect to existing PCDM systems and the evaluation of scalability and performance of PCDM systems, this review has identified directions in which PCDM research could invest more effort. These directions are discussed below with respect to existing PCDM systems and developing novel PCDM systems, respectively.

- Existing PCDM systems:
    - Perform experiments on scalability and performance with respect to growing traffic volumes.
    - Design experiments that demonstrate scalability and performance with respect to data complexity.
- Developing new PCDM systems:
    - Deploy relational databases, both row-oriented and column-oriented, in shared-nothing architecture and explore scalability and performance.
    - Explore avenues with NewSQL databases for PCDM in shared-nothing architecture. This includes exploring the suitability of graph databases, document databases, and key-value databases (NoSQL databases other than wide-column) for PCDM.

As previously stated, the scalability and performance of PCDM systems are mainly assessed with respect to the growing data volumes. In a few circumstances, parallel loading has been explored. Nevertheless, in developing highly scalable–data-intensive PCDM systems, performing experiments on other dimensions of scalability (i.e., traffic volume and data complexity) is equally vital. Similarly, as noted, the advent of new data models and the deployment of these novel data models in shared-nothing architecture, promise better scalability and performance. Thus, a pragmatic analysis in developing novel PCDM systems (e.g., by combining new data models and unexplored parallel architectures) is also crucial to future PCDM research.

Finally, emphasizing the need for ***an extensible, agile framework for methodical testing, evaluation, and the comparison of scalability and performances of heterogeneous PCDM systems*** are important. As noted in Section 6, as of today, there is no systematic approach to evaluate the scalability and performance of existing PCDM systems against each other. Thus, an agile extensible framework covering all parallel architectures and data models is vital. Such a framework is required to encompass characteristics of point cloud data so that different data sets can be directly compared. Furthermore, performance dimensions, such as storage performance, querying performance, and data loading performance,



should also be incorporated. We believe that such a framework requires careful analysis of the steps involved in PCDM experiments and data-intensive system characterizations.

## 8. Conclusions

LiDAR point cloud data are important sources for 3D geospatial scientific research. Currently, these inherently voluminous and heterogeneous data sets are being collected at unprecedented scales and densities. Today, there is a growing number of research attempts toward the development of highly scalable–performance-efficient PCDM solutions. These attempts are explored atop different parallel architectures and specific data models and have produced results that demonstrate scalability and commensurate performance.

With respect to the capacity, shared-memory architecture-based PCDM systems have been successfully explored by adopting databases that adhere to relational and NewSQL models. On the other hand, to date, there is only one system that has produced scalability and performance results with respect to managing point cloud data in a shared-disk architecture. Demonstrating scalability and performance results of shared-disk PCDM systems could provide important insights into PCDM research. More recently, there is a trend to adopt shared-nothing architecture and wide-column NoSQL
databases for PCDM systems.While shared-nothing architectures and wide-column models (and NoSQL models in general) are theoretically capable of achieving high performance and scalability, current results are insufficient to validate such claims with respect to PCDM.

The review of scalability and performance of parallel architectures and data models in PCDM systems and the evaluation of state-of-the-art PCDM systems presented herein aids in the identification of pivotal research gaps at the system level, as well as at the core research level. The main research gaps identified in this survey relate to gaps with respect to existing PCDM systems and gaps, with respect to developing novel PCDM systems. This work can be used as a guide in strategic decision-making when developing scalable PCDM systems under three main areas: (i) anticipated changes to point cloud data volume, (ii) anticipated changes in traffic volume, and (iii) other system requirements. Finally, this work has identified a critical need for an extensible, agile framework for methodical testing, evaluation, and comparison of scalability and performance of heterogeneous, PCDM systems that should be explored in future research.

**Author Contributions:** Conceptualization, C.N.L.H.; manuscript preparation, C.N.L.H., D.F.L. and A.-V.V.; manuscript reviewing and editing, D.F.L., N.-A.L.-K. and M.B.; supervision and funding, M.B. All authors have read and agreed to the published version of the manuscript.

**Funding:** This publication originated from research supported in part by a grant from Science Foundation Ireland under grant number SFI-17US3450. Further funding for this project was provided by the National Science Foundation as part of the project "UrbanARK: Assessment, Risk Management, & Knowledge for Coastal Flood Risk Management in Urban Areas" NSF Award 1826134, jointly funded with Science Foundation Ireland (SFI-17US3450) and the Northern Ireland Trust (grant USI 137).

**Institutional Review Board Statement:** Not applicable.

**Informed Consent Statement:** Not applicable.

**Data Availability Statement:** Not applicable.

**Conflicts of Interest:** The authors declare no conflict of interest.

## Abbreviations

The following abbreviations are used in this manuscript:

| | |
|---|---|
| 3D | three-dimensional |
| DBMS | database management system |
| DM | data management |
| FWF | full waveform |
| kNN | k nearest neighbor |
| LiDAR (LiDAR) | Light Detection and Ranging |
| OEDBM | Oracle Exadata Database Machine |
| PCDM | point cloud data management |
| PCs | point clouds |
| RDBMSs | relational database management systems |
| NoSQL | Not only SQL |
| SFC | space-filling curve |
| SFCs | space-filling curves |
| SQL | structured query language |

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
