# Peer review of "Scalability and Performance of LiDAR Point Cloud Data Management Systems: A State-of-the-Art Review"

_remotesensing, doi:10.3390/rs14205277_

Round 1
Reviewer 1 Report
This paper reviews the scalability and performance of PCDM systems in terms of parallel architectures and data models, and discussed their impact. In addition, a methodical approach for the selection of parallel architectures and data models for highly scalable and performance-efficient PCDM system development is proposed, along with some research gaps in the PCDM literature and possible directions for future research. In general, the paper is well-written. I think it can be published in Remote Sensing.
Author Response
Dear reviewer 1,
Thank you.
Please find the attached PDF. It synthesizes all the responses for comments received from all reviewers.
kind regards!

Reviewer 2 Report
See attached file.

Author Response
Dear reviewer 2,
Thank you very much. Your feedback helped us to improve our work significantly.
Please find the attached PDF and kindly check the responses we leave under reviewer #2. In addition to the responses to your comments, the PDF synthesizes all the responses for comments received from all reviewers.
kind regards!

Reviewer 3 Report
The manuscript presents a review of PCDM systems aimed at achieving higher scalability without comprimising performance. It compares the scalability and performance of the state-of-the-art PCDM systems based on lidar mapping in terms of architecture and data models. The work is interesting and useful for readers. However, I have some comments on the quality of its presentation listed below.
1. Figures A4 and A5 should be completely redrawn, I recommend to delete empty spaces and to increase the size of designations in them.
2. Tables 1 and 3 should be reformatted.
3. I also recommend to include the material of Appendix 1. Analysis of performance results and especiaally Table A5 that, to my mind, are very important, into the main body of the manuscript and discuss them in more details.
4. The abbreviation Structured Query Language (SQL) should be included in the list of abbreviations.
5. The term "LiDAR" in the text should be replaced by "lidar."
Based on the foregoing, I cannot recommend this manuscript for publication in its present form.
Author Response
Dear reviewer 3,
Thanks a great deal for your review and comments. Based on your reviews we were able to improve our work significantly.
Please find the attached PDF and check our responses for reviewer # 3. In the document, we've detailed how we address each comment. In addition, it synthesizes all the responses for comments received from all reviewers.
kind regards!

Reviewer 4 Report
The manuscript is relevant for the field but not clear presented. There are too many facts, and it is not clear where are the test results. Figures and tables properly show the data. The problem is important for LiDAR data storage and usage.
Author Response
Dear reviewer 4,
Many thanks for your comments. We improved our work significantly.
Please find the attached PDF and see our responses for reviewer #4. It also synthesizes all the responses for comments received from all reviewers.
kind regards!

Round 2
Reviewer 2 Report
I have not re-reviewed this article as I considered the original content complete and worthy of publication. The major change needed was careful editing by a conscientious English technical writer. Evaluating if the English writing is now acceptable is the responsibility of the Editor -- not a Reviewer.
If the Editor carefully reviews the English and decides it is acceptable, I recommend the article for publication.
In accord with this, I have retained my original comment that major English language changes are necessary, and I have marked all of the specific questions as "3-stars" -- i.e., a middle score -- except for the question about the quality of the English writing.
Author Response
Dear reviewer #2,
thanks for sharing your input on the revised version.
we'll be in touch with the editor.
your input indeed helped us to improve our work.
kind regards!